# Deficiency in FTSJ1 Affects Neuronal Plasticity in the Hippocampal Formation of Mice

**DOI:** 10.3390/biology11071011

**Published:** 2022-07-05

**Authors:** Viola von Bohlen und Halbach, Simone Venz, Simon Nwakor, Christian Hentschker, Elke Hammer, Heike Junker, Andreas W. Kuss, Oliver von Bohlen und Halbach, Lars R. Jensen

**Affiliations:** 1Institute of Anatomy and Cell Biology, University Medicine Greifswald, 17489 Greifswald, Germany; viola.bohlenundhalbach@uni-greifswald.de (V.v.B.u.H.); sinwak@t-online.de (S.N.); 2Department of Medical Biochemistry and Molecular Biology, University Medicine Greifswald, 17475 Greifswald, Germany; simone.venz@med.uni-greifswald.de (S.V.); hentschkec@uni-greifswald.de (C.H.); heike.junker@med.uni-greifswald.de (H.J.); 3Interfaculty Institute of Genetics and Functional Genomics, University Medicine Greifswald, 17475 Greifswald, Germany; hammer@uni-greifswald.de (E.H.); kussa@uni-greifswald.de (A.W.K.)

**Keywords:** FTSJ1, long-term potentiation, tRNA methyl transferase, neuronal plasticity, proteomics

## Abstract

**Simple Summary:**

Neuronal plasticity refers to the brain’s ability to adapt in response to activity-dependent changes. This process, among others, allows the brain to acquire memory or to compensate for a neurocognitive deficit. We analyzed adult FTSJ1-deficient mice in order to gain insight into the role of FTSJ1 in neuronal plasticity. These mice displayed alterations in the hippocampus (a brain structure that is involved in memory and learning, among other functions) e.g., in the form of changes in dendritic spines. Changes in dendritic spines are considered to represent a morphological hallmark of altered neuronal plasticity, and thus FTSJ1 deficiency might have a direct effect upon the capacity of the brain to adapt to plastic changes. Long-term potentiation (LTP) is an electrophysiological correlate of neuronal plasticity, and is related to learning and to processes attributed to memory. Here we show that LTP in FTSJ1-deficient mice is reduced, hinting at disturbed neuronal plasticity. These findings suggest that FTSJ1 deficiency has an impact on neuronal plasticity not only morphologically but also on the physiological level.

**Abstract:**

The role of the tRNA methyltransferase FTSJ1 in the brain is largely unknown. We analyzed whether FTSJ1-deficient mice (KO) displayed altered neuronal plasticity. We explored open field behavior (10 KO mice (aged 22–25 weeks)) and 11 age-matched control littermates (WT) and examined mean layer thickness (7 KO; 6 WT) and dendritic spines (5 KO; 5 WT) in the hippocampal area CA1 and the dentate gyrus. Furthermore, long-term potentiation (LTP) within area CA1 was investigated (5 KO; 5 WT), and mass spectrometry (MS) using CA1 tissue (2 each) was performed. Compared to controls, KO mice showed a significant reduction in the mean thickness of apical CA1 layers. Dendritic spine densities were also altered in KO mice. Stable LTP could be induced in the CA1 area of KO mice and remained stable at for at least 1 h, although at a lower level as compared to WTs, while MS data indicated differential abundance of several proteins, which play a role in neuronal plasticity. FTSJ1 has an impact on neuronal plasticity in the murine hippocampal area CA1 at the morphological and physiological levels, which, in conjunction with comparable changes in other cortical areas, might accumulate in disturbed learning and memory functions.

## 1. Introduction

The FTSJ1 (FtsJ RNA 2′-O-Methyltransferase 1) protein is expressed in different tissues, including the brain, where the hippocampus, the amygdala, and the corpus callosum in particular show pronounced expression [1]. It is involved in the 2′-O-methylation of the ribose moiety of nucleotides at positions 32 and 34 of the anticodon loop of its target tRNAs (see e.g., [2]). In humans, mutations in the X-chromosomal *FTSJ1* gene (FTSJ1, MIM #300499) have been repeatedly found to cause non-syndromic intellectual disability (ID) [1,3,4,5]. In order to gain more insight into the role of the genes that, in their mutated form, are involved in ID, different mouse models have been developed [6,7,8,9], including a gene-trap-based mouse model for FTSJ1-deficiency, which we established and characterized previously, to examine the functions of FTSJ1 in more detail. The FTSJ1-deficient mice were significantly smaller than wild-type littermates, and showed reduced body and bone mass. In addition, the mice showed increased corticosterone levels and hypoalgesia [10]. In our mouse model for FTSJ1 deficiency, gross morphological abnormalities in the brain were not observed, but an increased place-learning error rate [10] was noted, which hints at delayed or disturbed learning.

In 2021, another group established a different mouse model, deficient for FTSJ1, where young (8 week old) mutant mice displayed immature synaptic morphologies and aberrant synaptic plasticity, accompanied by behavioral and memory deficits [2]. However, it is unclear whether these phenotypes observed in juvenile FTSJ1-deficient mice persist into adulthood. In particular, mechanisms related to postnatal hippocampal formation could underlie further changes, since at the age of two months, the murine brain is not yet fully developed. Thus, there is still significant myelination occurring in the cortexes of mice between three and six months of age [11]. Moreover, dendritic spine densities in the hippocampal formation (area CA1 as well as the dentate gyrus) are significantly lower in 8-week-old mice as compared to 15-week-old mice of the same genotype [12]. It is possible that during the time from childhood to adolescence, the brain is reorganized by neuronal plasticity, leading to amelioration. However, it could also be the case that the repair mechanisms fail, and the effects seen in the young animals might be aggravated in adult mice. For that purpose, we examined, among other things, two correlates of neuronal plasticity in the hippocampal formation of adult *Ftsj1* knockout (KO) mice, a morphological one (changes in dendritic spines) and an electrophysiological one (long-term potentiation (LTP)). LTP was first described by Bliss and Lomo in the 1970s [13]. It is nowadays known to represent the major electrophysiological correlate for neuronal plasticity and is thought to subserve neuronal function associated with some forms of learning and memory [14,15,16]. LTP can be divided into two phases: (i) an early phase (E-LTP), which is independent of protein synthesis; and (ii) a late phase (L-LTP), which involves activation of transcription factors and depends on protein synthesis and can induce structural changes. These changes include, e.g., enlargement of the spines, which among other things involves actin polymerization [17]. Thus, LTP can promote the occurrence of multiple spine synapses, leading to a remodeling of existent dendritic spines and the generation of new dendritic spines [18,19,20]. Altered dendritic spines can be observed in different models of ID. Likewise, hippocampal LTP at CA1 synapses is altered in several animal models of ID. For example, mice expressing mutated *Mecp2* display impaired hippocampal LTP in area CA1 [21], as do *Fmr1* knockout mice [22]. In this study, we analyzed adult mice deficient in FTSJ1 to gain insight into the role of FTSJ1 in neuronal plasticity.

## 2. Materials and Methods

### 2.1. Mice

The mice were generated and kept as previously described [10]. In short, the *Ftsj1* gene-trapped mouse stem cell line (RRD143) was microinjected into blastocysts and subsequently used to generate chimeric pubs. Animals with germline transmission of the gene-trapped *Ftsj1* allele were backcrossed to C57BL/6J genetic background for >10 generations prior to phenotyping. The gene-trapped animals expressed no wildtype *Ftsj1* transcript [10]. The mice used for this study were between 22 and 25 weeks old at the time of testing.

### 2.2. Open Field

For open-field tests (OF), a quadratic 45 × 45 cm^2^ test arena was used (Panlab, Spain). Illumination was set to 25 lux. Mice (controls (WT): *n* = 11; FTSJ1^−/−^ (KO): *n* = 10) were placed in the middle of the area and allowed to explore for 7 min. Movements were recorded by a webcam (Logitech C300; Logitech, Zürich, Switzerland). Parameters characterizing open-field behavior were analyzed from recorded sessions using SmartJunior 1.0.0.7 (Panlab, Barcelona Spain). Between trials, the arena was cleaned with 70% ethanol.

### 2.3. Layer Thicknesses in Area CA1 and the Dentate Gyrus (DG)

Mice were euthanized and transcardially perfused with phosphate-buffered saline (PBS: pH 7.2: 10.73 g Na_2_HPO_4,_ 9.0 g NaCl and 2.0 g NaH_2_PO_4_ in 1 L distilled water) and afterwards with 4% paraformaldehyde (PFA (pH 7.2) dissolved in PBS,). Coronal sections (thickness: 30 µm) of the fixed brain (between Bregma −1.46 and −2.5) were made using a vibratome (VT 1000S, Leica, Germany) and collected in 20% ethanol. The sections were mounted on glass slides and then air-dried overnight. The next day, slices were incubated in sodium citrate buffer (pH 6.0) for 20 min using a microwave (800 W). After rinsing the sections, they were transferred into a phosphate-buffered saline (PBS) solution containing DAPI (0.1 µg/mL) in the presence of 3% serum and 0.1% Triton-X. Sections were rinsed again in PBS (3×) and embedded in Mowiol (Sigma-Aldrich, Taufkirchen, Germany). Images were acquired using an Olympus BX63 microscope fitted for fluorescence imaging. Images were analyzed using the software package cellSense Dimension (Olympus, Hamburg, Germany). The following layers were analyzed: (i) the basal layer of the hippocampal field CA1 (stratum oriens), (ii) the pyramidal layer of the hippocampal field CA1, (iii) the apical layer of the hippocampal field CA1 (stratum radiatum and lacunosum-moleculare), (iv) the upper molecular layer of the dentate gyrus (DG), and (v) the granular layer of the DG. The brains of adult FTSJ1-deficient mice (*n* = 7) and age-matched littermates (*n* = 6) were analyzed, and the observers were blinded to the groups under study. Per brain and structure, at least nine independent measurements were conducted.

### 2.4. Golgi Impregnation and Analysis of Dendritic Spines

Animals were euthanized and transcardially perfused with PBS, followed by perfusion with PFA. Brains were removed and immersed in the same fixative for at least five days. Golgi impregnations were performed according to the Golgi-Cox procedure (using Rapid GolgiStain reagents (FD NeuroTechnologies, Columbia, MD, USA)). Subsequently, slicesv120 µm thick (between Bregma −1.46 and −2.5) were made using a Leica VT 1000S vibratome (Leica, Wetzlar, Germany), mounted on gelatin-coated slides, and coverslipped with Merckoglas (Merck, Darmstadt, Germany). The three-dimensional reconstruction and analysis using the NeuroLucida system was conducted as recently described [23]. For each group, 5 brains were investigated, and the observers were blinded to the groups under study. In each case, about 20 individual dendrites were analyzed per region and brain. In all, 10,366 individual dendritic spines were reconstructed and analyzed in the basal layer of area CA1, 11,299 individual dendritic spines in the apical layer of area CA1, and 13,277 individual dendritic spines in the molecular layer of the upper leaf of the DG. The *n*-values for the statistical analysis were based on animal numbers and not on numbers of analyzed elements.

### 2.5. Electrophysiology

Acute horizontal brain slices were taken from adult mice and prepared using standard procedures, as described earlier [24]. ACSF-filled glass electrodes (1.5 mm borosilicate glass electrodes, with tip resistance of about 1–3 MΩ) were positioned in the stratum radiatum of the hippocampal field CA1 to record extracellular field potentials. Bipolar electrodes were used to stimulate the Schaffer collateral pathway. Field potentials were evoked by electrical stimuli (duration 100 ms) delivered at a frequency of 0.1 Hz. The stimulus intensity was adjusted to produce potentials of about 30% of the maximum amplitude. After recording a stable baseline for at least 15 min (single pulses at a 0.1 Hz rate), LTP induction was performed by a high frequent stimulation (HFS: 4 bursts, each 50 pulses at 100 Hz; interburst-interval: 300 ms). Data were collected using Signal 2.15 (Cambridge Electronic Design, Cambridge, UK) and transferred to Microsoft Office Excel 2003 (Microsoft, USA). Field potential amplitudes were averaged over six subsequent values each and specified as [% baseline].

### 2.6. Mass Spectrometry (MS)

The embedded sections that had been used for determining layer thicknesses in the hippocampus were transferred to xylene for several days in order to remove the coverslip. After removal of the coverslips, the sections (still attached to the slide) were rinsed in ethanol (96%, 70%, 50%, 20%; each for several minutes). Per animal, eight parts of area CA1 (around Bregma −2.06 to −2.30) containing the stratum oriens, stratum pyramidale, and stratum radiatum were dissected out using a scalpel and a binocular (Askania SMT4; Mikroskop Technik, Rathenow, Germany). The tissue collected from each animal was transferred into a tube containing 20% ethanol and stored. Dissected tissues were available for four mice samples (2 × wild type vs. 2 × FTSJ1-knockout), and were incubated in 40 µL extraction buffer (100 µL 4% SDS, 100 mM DTT in 100 mM TRIS pH 8.0) at 4 °C for 5 min, before protein extraction at 100 °C for 20 min and subsequent shaking for 2 h at 80 °C in a thermomixer and inversion of samples every 10 min. After cooling on ice for 1 min, residual particles were pelleted by centrifugation (14,000× *g*, 15 min) and the protein containing supernatant was collected and stored at −80 °C.

For MS analysis, peptide lysates were prepared following the adapted SP3 protocol (single-pot solid-phase enhanced sample preparation) using hydrophobic and hydrophilic Sera-Mag SpeedBeadsTM (Cytiva, Marlborough, MA, USA) as described [25,26]. The protein samples were diluted with 40 µL 20 mM Tris-HCl and subjected to reduction (25 mM dithiothreitol (DTT) at 37 °C, 30 min) and alkylation (100 mM iodoacetamide at 37 °C, 15 min). Alkylation reaction was quenched by addition of 25 mM DTT. Subsequent protein digestion with trypsin (enzyme to protein ratio 1:25) and purification of peptides was performed on SP3 beads according to [27]. Peptides were analyzed on a QExactive HF Hybrid Quadrupole-Orbitrap Mass Spectrometer (Thermo Scientific, Bremen, Germany) coupled to a nano-LC system (Ultimate 3000, Thermo Scientific, Bremen, Germany). Mass spectra were recorded in data-independent acquisition (DIA) mode, and data were analyzed via the DirectDIA algorithm implemented in Spectronaut (Biognosys, Zurich, Switzerland) using a Uniprot/SwissProt database (v. 2021_02) for mouse sequences. Proteins were only selected for further analyses if at least two unique + razor peptides were identified and quantified per protein. A median normalization was performed at the ion level before connecting the statistical analysis on peptide level. Binary differences were identified by application of a reproducibility-optimized test statistic (using the ROTS package). Only proteins that showed different abundance (*p*-value < 0.05) were used for further considerations. Detailed descriptions of data acquisition and search parameters are provided in the Appendix A.

Assignment of proteins to functions and pathway enrichment analysis was performed using Ingenuity pathway analysis (IPA, Qiagen, Hilden, Germany).

### 2.7. Statistical Analysis

Prism 5 for Windows (GraphPad Software Inc., San Diego, CA, USA) was used for statistical analyses. The Shapiro–Wilk test was used to analyze for normal distribution. Thereafter, for statistical evaluation, either Mann–Whitney U-tests (behavior, morphological analyses) or two-sided *t*-tests (electrophysiology, MS) were used. Significance levels for all tests were set to *p* ≤ 0.05; significant changes in the figures were indicated by an asterisk.

## 3. Results

### 3.1. Open Field

In the open field test (OF), it was first analyzed whether the FTSJ1-deficient mice displayed abnormal motor behavior. As compared to age-matched littermates (WT), the FTSJ1-deficient mice (KO) moved about the same distance and with nearly the same mean velocity (Figure 1a,b). The FTSJ1-deficient mice spent less time in the center of the arena as compared to the controls; however, this effect did not reach significance (FTSJ1^−/−^ mice: 52.44 s versus controls: 35.01 s; *p* = 0.061; Figure 1c).

### 3.2. Thickness of Layers in the Hippocampal Area CA1 and in the Dentate Gyrus (DG)

We first analyzed the gross morphology of the brains of adult FTSJ1-deficient mice and their age-matched control littermates, but did not find obvious alterations. Next, the hippocampal formation was analyzed in detail, focusing on the hippocampal field CA1 (Figure 2a,b) and the DG (Figure 2a,c). The thickness of the basal layer, as well as that of the pyramidal layer, was not different between the groups, but the thickness of the apical layer was significantly reduced in FTSJ1-deficient mice as compared to controls (384.7 ± 6.33 µm vs. 407.1 ± 7.57 µm; *p* = 0.042; Figure 2b). Concerning the dentate gyrus, neither the thicknesses of the molecular layer nor of the granular layer were different between the two groups (Figure 2c).

### 3.3. Dendritic Spines in the Hippocampal Area CA1 and in the DG

Since changes in neuronal plasticity are linked to learning and memory, we analyzed a morphological hallmark of neuronal plasticity, namely, changes in dendritic spines. Within the hippocampal area CA1, we analyzed dendritic spines of basal dendrites of pyramidal neurons (located in the stratum oriens) as well as dendritic spines of apical dendrites of CA1 pyramidal neurons that can be found in the stratum radiatum and the stratum lacunosum-moleculare (Figure 3a). In addition to that, dendritic spines were analyzed in the molecular layer of the upper leaf of the dentate gyrus (Figure 3a). These dendritic spines belong to the dendrites of granule cells located in the granular layer. Alterations in dendritic spines were only seen in the apical dendrites of CA1 pyramidal neurons. The density of dendritic spines of apical CA1 neurons of FTSJ1 knockout mice was significantly higher than those of the control littermates (FTSJ1 knockout: 2.003 ± 0.04 versus controls: 1.693 ± 0.07; *p* = 0.032; Figure 3b). The dendritic spines of the basal dendrites of CA1 pyramidal neurons seemed to be unaltered, as were the dendritic spines of the granule cells of the DG (Figure 3c,d).

### 3.4. LTP within Hippocampal Area CA1

LTP, an electrophysiological correlate of neuronal plasticity, was analyzed in area CA1 of adult FTSJ1-deficient mice in comparison to age-matched control littermates. In both adult FTSJ1-deficient mice and in the respective controls, stable LTP could be induced and could be recorded over a prolonged period (Figure 4a). However, shortly after induction, the LTP in mice lacking FTSJ1 was slightly weaker than the LTP seen in controls. Thus, about half an hour after LTP induction, LTP in FTSJ1-deficient mice was significantly weaker as compared to controls (20–30 min after LTP induction; Figure 4a,b). LTP in the FTSJ1-deficient mice remained stable even one hour after LTP-induction; however, LTP was significantly lower (Figure 4c).

### 3.5. Mass Spectrometry

In order to gain an impression with respect to molecules that are altered in the adult hippocampal formation, mass spectroscopy was performed focusing on CA1 tissue from adult FTSJ1-deficient mice and age-matched control littermates. In total, 2557 different proteins were identified, and 166 proteins showed a differential abundance (*p*-value < 0.05) predominantly at small-fold differences. Twenty-seven proteins were found with a fold difference of 1.3 (higher levels in KO: 19; lower levels in KO: 8; Appendix A). The highest upregulation was seen in the FXYD domain containing ion transport regulator 6, which is encoded by the Fxyd6 gene. Fxyd6 is, among others, involved in modulating Na/K pump functions [28]. A closer inspection of the differentially abundant proteins, using IPA with a focus on database content related to the central nervous system and cell types located in the brain, disclosed significant enrichment of these proteins in seven categories of “neurological diseases” and “nervous system development and function” (Table 1). The interaction of the proteins is visualized in the network “nervous system development and function, cell death and survival, tissue morphology” (Figure 5). In particular, the increase in myelin-associated glycoprotein (MAG) and myelin basic protein (MBP) fits the differences found for dendritic spines in the CA1 region by microscopy. Direct and indirect interactions point to a function of serine/threonine-protein kinase PRKC in this network, but the abundance of the subunits detected by mass spectrometry did not differ. Another central molecule in this network is brain-derived neurotrophic factor (BDNF). However, the relevance of BDNF in FTSJ1-deficient animals must be analyzed in detail, and could be in the focus of future studies. Additionally, in the hippocampus of FTSJ1-deficient mice, higher levels of annexin A2, also known as annexin II (ANXA2), were observed. This protein is thought to be involved in hippocampus-dependent learning [29]. Furthermore, several other proteins involved in neuronal functions, such as nectin 1 (NECT1), were found to be upregulated.

## 4. Discussion

Adult FTSJ1-deficient mice survived into adulthood and did not display disturbances in motor behavior, since they navigated through the open field arena in a fashion comparable to that of the age-matched littermate controls, indicating that the mutant animals have no deficits in motor behavior. This is also in keeping with the behavior seen in 8-week-old FTSJ1-deficient mice [2]. Somewhat comparable to the 8-week-old FTSJ1-deficient mice, the adult mutants spent less time in the center of the open field.

Likewise, the brains of the adult FTSJ1-deficient animals did not show gross morphological changes. A more detailed analysis revealed that within their hippocampi, morphological differences could be seen in area CA1. This area plays an important role in learning and memory, as it contributes to the acquisition of context-dependent extinction and is required for contextual memory retrieval [30]. The hippocampal area CA1 is well-known for its capacity for neuronal plasticity, and LTP is among the best-investigated phenomena in this part of the brain [31]. LTP is the most popular form of synaptic plastic change that can occur during learning [32], and different mouse models of IDs showed altered LTP (e.g., [21,22]). This is in line with the previous finding that young FTSJ1-deficient mice show a strongly impaired LTP in the hippocampus, which could be induced, but with a reduced amplitude of about 50% [2]. Likewise, LTP can be induced in the hippocampal area CA1 of adult FTSJ1-deficient mice, and remains stable over time, as we show in this report. As compared to age-matched controls, the difference in LTP that we observed in adult mutant animals was not as prominent as reported for the young FTSJ1-deficient mice. This might be due to the ability of the hippocampal formation to adapt via neuronal plasticity. In other words, hippocampal neuronal plasticity might be able to compensate for certain defects that result in delayed postnatal development.

Young (8 weeks of age) FTSJ1-deficient mice display an overall very strong reduction (about 30%) in spine densities in the hippocampal formation and cortex as compared to wild-type animals [2]. Concerning this issue, it should be kept in mind that at this age at least, the hippocampal formation is not fully interconnected, and numbers of dendritic spines (in area CA1 as well as in the DG) are significantly lower at an age of about 2 months, as compared, for example, to an age of about 4 months [12]. It is possible that the effects on dendritic spines seen in the young FTSJ1-deficient mice persist into adulthood. Our results indicate that postnatal spinogenesis is delayed in mutant mice, since the dendritic spine densities of DG neurons and the spine densities of basal dendrites of CA1 pyramidal neurons were not different from age-matched controls. Concerning the DG, it should be noted that this brain region is capable of adult neurogenesis that also can contribute to neuronal plasticity as well as to learning and memory. Thus, it is possible that the connectivity of the hippocampal formation could also be altered at this level, and that the effects seen in the young FTSJ1-deficient mice differ from those of adult mutants, since the rate of adult neurogenesis in the postnatal dentate gyrus is different between young and adult mice [33]. Importantly, dendritic spine densities of the apical CA1 dendrites are altered in the FTSJ1-deficient mice, indicating a disturbance in neuronal plasticity. Within the hippocampus, we found a significant upregulation of FXYD6. Mutations in *Fxyd6* have been associated with schizophrenia [34,35]. FXYD6 has been identified as a synaptic protein [36] and is involved, for example, in modulating Na^+^/K^+^-ATPase. Na+, K (+)-ATPase activity can, for example, be observed during the development of LTP [37] and thus might contribute (e.g., as a compensating factor) to neuronal plasticity in the adult FTSJ1-deficient mice. The next gene product that was found to be upregulated in the hippocampal formation of FTSJ1 knockout mice was annexin A2, which is associated with lipid rafts localized in neuronal dendrites of hippocampal pyramidal neurons, and seems to be involved in hippocampus-dependent learning and memory [29]. A third protein that was found to be upregulated was nectin-1, an immunoglobulin-like adhesion molecule that is involved in synapse formation within the hippocampal formation [38]. Concerning area CA3 of the hippocampal formation, it has been shown that nectin-1 regulates neuronal activity by suppressing the GABAergic inhibitory synaptic transmission [39]. Furthermore, myelin-associated glycoprotein (MAG) and myelin basic protein (MBP), factors that are also known to be differentially regulated in different types of learning [40,41], were also found to be upregulated. Myelin insulates nerve cells and increases the speed of the electrical impulses that travel along the axons. As proper myelination is important for sensory function, it is interesting to speculate that increased pain tolerance, as observed in FTSJ1-deficient mice [10], could be related to defective myelination in these mice. Likewise, differential expression of GAP43, also termed neuromodulin, was detected. GAP43 is involved in a variety of different functions in the nervous system, such as neuronal regeneration, signal transduction, synaptic potentiation [42] and axonal plasticity [43]. Interestingly, our network analysis highlighted BDNF as a molecule in this network. BDNF is a key molecule involved in neuronal plasticity and in learning and memory. Among other actions, lack of BDNF impairs hippocampal LTP [44] and impairs synaptic formation in the DG [45]. Additionally, BDNF plays an important role in adult neurogenesis in the hippocampus [46].

Whether these alterations are directly linked to FTSJ1 deficiency or to plasticity will be an important issue for future investigations, since it is possible that these alterations try to compensate or counteract the deleterious effects of a loss of FTSJ1, as suggested by the fact that the effects on neuronal plasticity and behavior were much more dramatic in juvenile FTSJ1-deficient mice at the age of 8 weeks [2] as compared to the adult (20–25 weeks of age) knockout mice we investigated here. However, we are able to present a combination of electrophysiology and proteomics data from a very defined anatomical area, and the relatively small divergence we found between adult mutant and WT mice in our dataset indicates that the observed alterations in spine density and LTP might not be due to FTSJ1dependent deregulation of individual proteins in the cells from the investigated tissue. Our findings rather point to an FTSJ1-related disturbance in a delicate and finely tuned subcellular interplay of proteins and lipids that forms the molecular basis for the cognitive aspects of brain function. 

## 5. Conclusions

We were able to show that FTSJ1 deficiency affects neuronal plasticity in the hippocampus of adult mice on the electrophysiological as well as on the morphological level. The proteomic analysis and subsequent network analysis allowed the assignment to affected signaling pathways, indicating that compensatory mechanisms may ameliorate the effects induced by FTSJ1 deficiency in adult mice.

## Figures and Tables

**Figure 1 biology-11-01011-f001:**
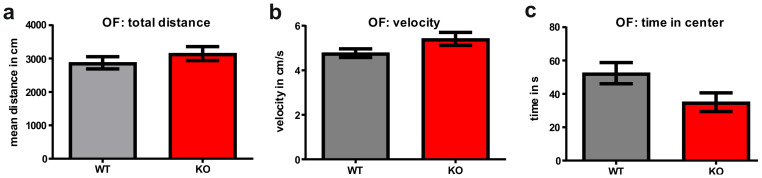
Behavior of FTSJ1-deficient mice (KO) and age-matched littermates (WT) in the open field (OF) test: mice of both genotypes explored the OF arena. (**a**) Within a given time, the mouse types of did not differ concerning the distance they moved. (**b**) FTSJ1-deficient mice showed comparable velocity when moving through the open field arena. (**c**) The FTSJ1-deficient mice spent less time in the center of the OF as compared to their controls. Despite this obvious effect, the difference was not statistically significant (*p* = 0.061). Data are expressed as mean ± SEM.

**Figure 2 biology-11-01011-f002:**
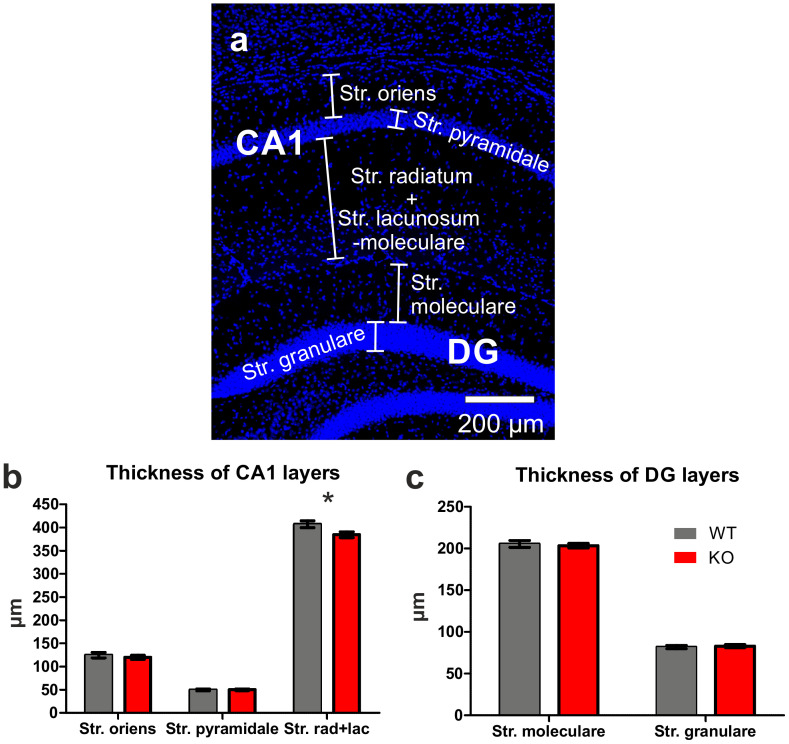
Thickness of different hippocampal layers. (**a**) Serial coronal sections counterstained with DAPI were used for the analysis of the thickness of different layers of the hippocampal area CA1, as well as of the dentate gyrus (DG). (**b**) Analysis of the different layers of the hippocampal area revealed a significant reduction in mean size (*p* = 0.042) only in the case of the stratum radiatum and stratum lacunosum-moleculare, while in the other layers, no significant changes in the mean thickness were seen. (**c**) Analysis of mean thicknesses of the molecular and granular layers of the upper leaf of the dentate gyrus revealed no significant differences between age-matched FTSJ1-deficient mice and their control littermates. Data are expressed as mean ± SEM. * indicates significant differences (*p* < 0.05).

**Figure 3 biology-11-01011-f003:**
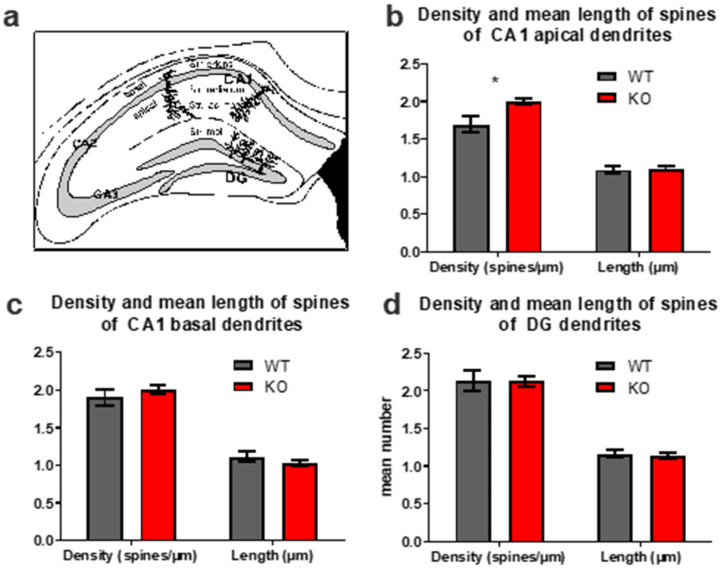
Analysis of dendritic spines. (**a**) Schematic drawing of a hippocampal section indicating the positions of the apical and basal dendrites of CA1 pyramidal neurons and of the dendrites of the granule cells of the upper leaf of the DG that extends into the molecular layer. (**b**) Analysis of dendritic spines of the apical dendrites of Area CA1 revealed that the mean length of dendritic spines was not different between FTSJ1-deficient mice and the respective controls, but that spine densities were significantly increased in FTSJ1-deficient mice. (**c**) Analysis of dendritic spines on the basal dendrites of CA1 pyramidal neurons revealed no differences in either mean spine densities or mean spine length. (**d**) Dendritic spines in the molecular layer of the dentate gyrus did not differ between FTSJ1 knockout and control mice. Data are shown as mean ± SEM. * indicates significant differences (*p* < 0.05).

**Figure 4 biology-11-01011-f004:**
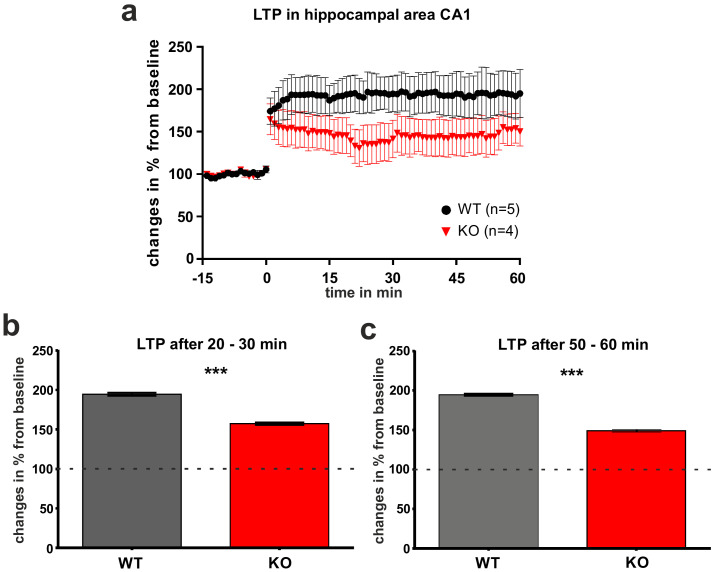
LTP within hippocampal area CA1. (**a**) LTP could be induced in both controls and FTSJ1-deficient mice, and stable LTP could be recorded over a long period. (**b**) About half an hour after LTP induction, LTP in FTSJ1-deficient mice (KO) was weaker as compared to controls (WT) and differed significantly from controls (20–30 min after LTP induction). (**c**) About one hour after LTP induction, the FTSJ1-deficient mice (KO) still displayed a stable LTP as compared to controls (WT), but it remained at a lower level. Dotted line indicates baseline level set to 100%. Data are shown as mean ± SEM; baseline was set to 100%. *** indicates significant differences (*p* < 0.001).

**Figure 5 biology-11-01011-f005:**
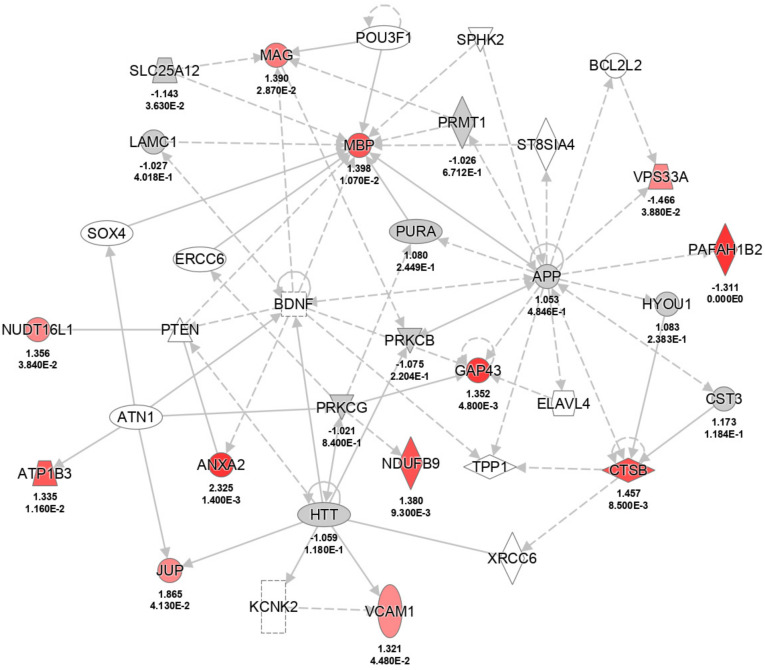
Ingenuity pathway analysis (IPA)-visualization of the network “nervous system development and function, cell death and survival, tissue morphology”. Proteins fitting the parameter *p* < 0.05 and fold change of 1.3 are highlighted in different shades of red according to the significance (*p*-value) of the difference between WT mice and mutant mice.

**Table 1 biology-11-01011-t001:** Enrichment pathway analysis results (IPA)—diseases and functions. The analysis was narrowed down to molecules related to tissue and cell lines of the central nervous system. Twenty-seven significantly different proteins (*p* < 0.05, FC > |1.3|) were analyzed and categories with a significant enrichment of proteins are shown. As supportive data, all molecules with *p* < 0.05 are provided in column 5. The proteins, which showed a fold change of 1.3, are highlighted in italics and bold.

Categories	Diseases or Functions Annotation	*p*-Value	Molecules *p* < 0.05; FC 1.3	Molecules *p* < 0.05
Neurological Disease	Hypomyelination of axons	7.84 × 10^−5^	MAG, MBP	***MAG***, ***MBP***
Neurological Disease, Organismal Injury and Abnormalities	Progressive encephalopathy	3.70 × 10^−3^	ANXA2, GAP43, MAG, MBP	ALDH1L1, ***ANXA2***, ATP5F1A, ATP8A1, ENO2, FTH1, GAK, ***GAP43***, GNAS, ***MAG***, ***MBP***, NAE1, TUBA1A
Cellular Development, Cellular Growth and Proliferation, Nervous System Development and Function, Tissue Development	Outgrowth of neurites	4.85 × 10^−3^	GAP43, MAG, RAB22A, VCAM1	BASP1, ***GAP43***, GNAS, HRAS, ***MAG***, PLXNA4, PRKACA, ***RAB22A***, SLC25A5, SRC, SYN1, TUBA1A, ***VCAM1***
Cell Morphology, Cellular Assembly and Organization	Elongation of cellular protrusions	8.21 × 10^−3^	GAP43, MAG	ALCAM, ***GAP43***, GNAS, ***MAG***
Cell Death and Survival, Neurological Disease, Organismal Injury and Abnormalities	Cell death of hippocampal neurons	1.03 × 10^−3^	ATP2C1, VPS33A	***ATP2C1***, CAT, GSK3A, KSR1, NAE1, PLXNA4, SLC25A12, SRC, ***VPS33A***
Neurological Disease, Skeletal and Muscular Disorders	Neuromuscular disease	2.58 × 10^−2^	ANXA2, GAP43, MBP, NDUFB9	ALCAM, ***ANXA2***, AP1S1, ATP5F1A, ATP5F1B, BASP1, ENO2, FKBP4, FTH1, ***GAP43***, GNAS, MAP2K4, ***MBP***, ***NDUFB9***, NDUFS6, Tmsb4x, TUBA1A, UQCRC2
Nervous System Development and Function, Neurological Disease	Abnormal morphology of central nervous system	2.59 × 10^−2^	CTSB, GAP43, MAG, MBP	

## Data Availability

Data available on request due to restrictions. The data presented in this study are available on request from the corresponding author (OvB).

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
