# Peer review of "Deficiency in FTSJ1 Affects Neuronal Plasticity in the Hippocampal Formation of Mice"

_biology, 2022, doi:10.3390/biology11071011_

Round 1

Reviewer 1 Report

The authors of this study employed an animal model for assessing how Ftsj1 gene promote intellectual disabilities. Concretely, they studied how this gene might affect brain plasticity, long-term potentiation (LTP) in the CA1 hippocampal area. As many preclinical studies, this can guide future studies conducted with humans. Anyway, it should be necessary to clarify certain aspects to improve the quality of the study.

Abstract:

It should be necessary to clarify in abstract how this gene might be linked with intellectual disabilities as they only stated their link with memory process through alterations in hippocampus, which is closely related with memory process instead of the IQ. Therefore, it should be completely necessary to provide enough background for reinforcing validity of this model.

They did no provide information regarding the number of subjects that they have been employed in the study. That is, clarifying and providing methodological Information.

The conclusions derived from the study were far from clarifying their involvement for intellectual disabilities and clinical applications.

Introduction:

Intellectual disabilities are a broad construct that did not offer to much information regarding the main purpose of the study. That is, an IQ bellow 70 was common for many mental disorders, but this not necessarily correspond with morphological brain abnormalities. In this sense, not only is necessary assessing morphological alterations, but also functional connectivity. This fact diminished my interest for the study as it was necessary to reinforce or clarify how this animal model clarify this “intellectual disability”.

Method:

 Authors did not provide information regarding how they calculated the number of subjects that they need for conducting this study. I think that this is essential for conducting an empirical research study.

I would like to know whether authors assessed whether data was normally distributed or not. Based on this, it would be necessary to applied non-parametric analysis or log transformed data.

Due to the high number of comparisons between groups and the exploratory nature of this study, it would be necessary to apply same statistical control for reducing the type 1 error. For example, by applying the Bonferroni correction.

According to these comments, this would change conclusions derived from previous results. Furthermore, it should be properly discussed clinical applications of this study as stated in the introduction.

Author Response

We wish to thank the reviewers for their helpful comments and advices. Based on their suggestions we have tried to clarify several aspects in order to improve the manuscript.

Rev #1

We wish to thank the reviewers for their helpful comments and advices. Based on their suggestions we have tried to clarify several aspects in order to improve the manuscript.

Abstract
As the title of our manuscript indicates, we analyzed neuronal plasticity within the hippocampus of FTSJ1 deficient mice. To void any confusion, we re-wrote the abstract and the introduction to clarify that the topic of that manuscript is not on intellectual disability (ID)but on neuronal plasticity.
We have included a sentence in the abstract: “As FTSJ1 is involved in protein synthesis through modification of certain tRNAs we performed a …” describing the importance of FTSJ1 in protein synthesis and thereby providing a link to the observed phenotype. There is good evidence for FTSJ1 being involved in protein synthesis, but whether this is also involved in the ID phenotype can not be answered yet. We would prefer to be very careful about statements on the molecular role of FTSJ1 in ID in the abstract.

We have not indicated the number of subjects used in this study in the abstract. We have not investigated human subject. In this study mice (FTSJ1 deficient mice and their controls) were used. The number of animals used for each experiment are indicated in the manuscript, e.g. in the section Materials & Methods, subchapter 2.2, 2.3, 2.4 and 2.6 . The number of animals used in case of the LTP experiments are indicated in figure 4.
“The conclusions derived from the study were far from clarifying their involvement for intellectual disabilities and clinical applications” Unfortunately, the results of our study can not be used to make a recommendation for a clinical application.
Introduction:
We have re-written the introduction by focusing on neuronal plasticity.

Method:
We have not included subjects in an empirical study. We have used mice for that research study. As suggested we now use the Mann-Whitney U-test for the analysis of the behavioral data as well as for the electrophysiological and morphological data.

We are not sure what the reviewers means with “due to the high number of comparisons between groups and the exploratory nature of the study”. We have only made pair-wise comparisons and analyzed data that were obtained by measurements. 

Based on our results, a clinical application of that study is not stated in the introduction.
Moreover, the data that we obtained were generated by analyzing alterations in neuronal plasticity in the hippocampus of FTSJ1 deficient mice. With these data and the data that have been published by others on that topic, a suggestion for a clinical application is not suitable.

Reviewer 2 Report

This is an interesting study showing as a mutation in the FTSJ1 gene has an impact on neuronal plasticity in the hippocampal formation of mice.

The reviewer has some comments and suggestions.

1 – Use “hippocampus” or “hippocampal formation”. Do not use both, although the reviewer prefers “hippocampal formation”.

2 – Regarding the quantification, it is important to know if the observers were blinded to the groups under study.

3 – The hippocampal formation is a very complex structure. Which part was studied? Were the levels matched for comparison when studying the layers and dendrites? A photograph is essential to evaluate the quality of the material.

4 – Present the reason for studying only the upper leaf of the dentate gyrus.

5 – Again, photographs should be present to evaluate the quality of the Golgi material.

6 –How did the authors perform the selection of neurons to be selected for the spine counting? Was it an unbiased selection or were there criteria for selection?

7 – Although present in reference 25, the reviewer prefers a little more detail in the methodology of the quantification of spine parameters. Also, an image would be helpful for the reader to understand the methodology used.

8 – BDNF is a key molecule involved in learning, memory and neurogenesis. Some more discussion concerning BDNF is necessary in the Discussion section.

Author Response

We wish to thank the reviewers for their helpful comments and advices. Based on their suggestions we have tried to clarify several aspects in order to improve the manuscript.

1    As suggested “hippocampus” has been replaced by “hippocampal formation” throughout the whole manuscript.

2    We added that the observers were blinded to the groups under study in the section: “Layer thicknesses in area CA1 and the dentate gyrus (DG)” and “Golgi impregnation and analysis of dendritic spines”

3    In this study, the hippocampal area CA1 and the dentate gyrus were analyzed (between Bregma -1.46 and -2.5). In area CA1, dendrites in the apical and basal layer were analyzed separately. In the dentate gyrus only dendrites (of granule cells located in the upper leaf of the dentate gyrus) in the molecular layer were analyzed. Individual dendritic parts were analyzed under high-power magnification and reconstructed using NeuroLucida. Since reconstructions were made in a 3-dimensional space /the volume of the section) and not in a single plane (as e.g. by analyzing single images), no images were taken. Using a magnification of about 1000x only parts of a dendrite will be visible in a photograph (or a single plane) and a photograph of the whole section in this high magnification is not possible. Since we have indeed analyzed a part of area CA1 and the DG, we added to the chapters “Layer thicknesses in area CA1 and the dentate gyrus (DG)” and “Golgi impregnation and analysis of dendritic spines” that the analysis was done using the area around Bregma -1.46 and -2.5

4    Only the upper leaf of the DG has been analyzed since the upper and lower leaf differ in their morphology. For example, the molecular layer of the lower leaf is thinner as the molecular layer of the upper leaf. In addition, the morphology of the granule cells in the upper and lower leaf are somewhat different (e.g. (Chacon et al. 2003). In order to reduce the variance of the individual neurons only the upper leaf was analyzed.  

5-7. The journal has made a “similarity check” and has found that our part describing the method of Golgi impregnation and the subsequent analysis of dendritic spines is a sort of a “plagiarism” of former published papers of us and we were advised to reduce these similarities in this manuscript. Therefore, the description of the methods has been shortened and we only cite one of the papers in which the method is described in detail. Further descriptions can be found e.g. in (von Bohlen und Halbach et al. 2006; Baum et al. 2016; Bertram et al. 2016; Bracke et al. 2018; Bracke et al. 2019; Schepers et al. 2020).
Concerning the selection of neurons for dendritic spine counting: 
We have not selected specific neurons. We have selected dendrites located in the appropriate region. The dendrites were randomly selected and were analyzed if they met the following parameters: Only dendrites were evaluated that displayed no breaks in their staining (Leuner et al. 2003) and that were not obscured by other structures or artifacts (Liu et al. 2001). Moreover, only spines located on secondary or tertiary dendritic trees were evaluated (this is due to the fact that primary dendrites have low spine densities and since the primary dendrites are very thick -> thus, most of the dendritic spines are “hidden”). In addition, only one segment per individual dendritic branch was chosen for the reconstruction. Sampling was optimized to produce a coefficient of error (CE) under the observed biological variance.
Concerning the three dimensional reconstruction and the methodology there are two papers describing the technique of the NeuroLucida system in detail (Glaser and Glaser 1990; Dickstein et al. 2016); however, Dickstein and coworkers used NeuroLucida 360, which is a little bit different from the “pure” NeuroLucida-System.

8. Indeed, BDNF is a key molecule involved in learning, memory, neuronal plasticity and adult neurogenesis. We have now added a short paragraph in the discussion concerning the role of BDNF in the hippocampus.

References:

Baum P, Vogt MA, Gass P, Unsicker K, von Bohlen und Halbach O (2016). Fgf-2 deficiency causes dysregulation of arhgef6 and downstream targets in the cerebral cortex accompanied by altered neurite outgrowth and dendritic spine morphology. Int J Dev Neurosci 50, 55-64
Bertram J, Koschutzke L, Pfannmoller JP, Esche J, van Diepen L, Kuss AW, Hartmann B, Bartsch D, Lotze M, von Bohlen und Halbach O (2016) Morphological and behavioral characterization of adult mice deficient for srgap3. Cell Tissue Res, vol 366, pp 1-11
Bracke A, Domanska G, Bracke K, Harzsch S, van den Brandt J, Broker B, von Bohlen und Halbach O (2019). Obesity impairs mobility and adult hippocampal neurogenesis. Journal of Experimental Neuroscience 13, 
Bracke A, Schafer S, von Bohlen und Halbach V, Klempin F, Bente K, Bracke K, Staar D, van den Brandt J, Harzsch S, Bader M, Wenzel UO, Peters J, von Bohlen und Halbach O (2018). Atp6ap2 over-expression causes morphological alterations in the hippocampus and in hippocampus-related behaviour. Brain Struct Funct 223, 2287 - 2302
Chacon MA, Reyes AE, Inestrosa NC (2003). Acetylcholinesterase induces neuronal cell loss, astrocyte hypertrophy and behavioral deficits in mammalian hippocampus. J Neurochem 87, 195-204
Dickstein DL, Dickstein DR, Janssen WGM, Hof PR, Glaser JR, Rodriguez A, O'Connor N, Angstman P, Tappan SJ (2016). Automatic dendritic spine quantification from confocal data with neurolucida 360. Curr Protoc Neurosci 77, 1 27 21-21 27 21
Glaser JR, Glaser EM (1990). Neuron imaging with neurolucida--a pc-based system for image combining microscopy. Comput Med Imaging Graph 14, 307-317
Leuner B, Falduto J, Shors TJ (2003). Associative memory formation increases the observation of dendritic spines in the hippocampus. J Neurosci 23, 659-665
Liu WS, Pesold C, Rodriguez MA, Carboni G, Auta J, Lacor P, Larson J, Condie BG, Guidotti A, Costa E (2001). Down-regulation of dendritic spine and glutamic acid decarboxylase 67 expressions in the reelin haploinsufficient heterozygous reeler mouse. Proc Natl Acad Sci U S A 98, 3477-3482
Schepers J, Gebhardt C, Bracke A, Eiffler I, von Bohlen und Halbach O (2020). Structural and functional consequences in the amygdala of leptin-deficient mice. Cell Tissue Res 382, 421-426
von Bohlen und Halbach O, Zacher C, Gass P, Unsicker K (2006). Age-related alterations in hippocampal spines and deficiencies in spatial memory in mice. J Neurosci Res 83, 525-531

Round 2

Reviewer 1 Report

I'm somewhat confused by the answers that have been provided by the subjects of my comments. In this sense, they have not solved the problems that I pointed out above.

For example, I asked them to structure the abstract. In this sense, they would need to put an introduction, objectives, methodology, results, and conclusions.

In fact, even if it is research with animals (subjects), it is necessary to specify the methodology in the abstract.

If the authors doubt the long-term application of their research. That is, to eliminate that information from the abstract, the introduction and the discussion.

Therefore, I would appreciate it if you were more explicit about the usefulness of your results and if you did not extrapolate to human. Please remove this information.

I don't know if you have understood the comment I made about statistical analyses. In this sense, although they have used animals, the statistical analyzes are the same. In fact, a very low number of subjects and a high number of comparisons considerably increases the type 1 error. Therefore, it is necessary to apply a correction factor. They must also check if the data conform to normality and, based on this, apply the subsequent statistical analysis.

It would be appropriate for them to remove any comments on the applications of their human study and stick to the results.

Author Response

We wish to thank the reviewer for the comments and suggestions. Indeed, we have sometimes some difficulties to understand what the reviewer means in detail. 
We are sorry for that. We hope that we could explain the obscurities we came across.

“I'm somewhat confused by the answers that have been provided by the subjects of my comments. In this sense, they have not solved the problems that I pointed out above”.
Indeed, there has been a misunderstanding regarding the comments of reviewer #1 in the first revision. Now we understand that the reviewer asked us to structure the abstract into different subsections. 
Now, we have revised the abstract and structured it into sections (objective, methods, results, and conclusions) and as requested in the former comments of the reviewer, we added the number of animals used for each method. As outlined in the “guide for Authors” the abstract is a single paragraph that follows the style of the structured abstracts, but without headings. Furthermore, the abstract should below or equal 200 words; the current abstract has 199 words.

We do not really understand what the reviewer means by “the authors doubt the long-term application of their research. That is, to eliminate that information from the abstract, the introduction and the discussion“ 
We have no doubt that the methods we used are state-of-the-art techniques in electrophysiology and neuroanatomy. We have no doubt that these techniques will be applied also in the future; not only by us, but also by other groups and will create robust results.
The groundbreaking discovery of LTP has been in the last century by Bliss and Lomo (Bliss and Lomo 1973; Bliss and Collingridge 1993) and the role of  LTP for synaptic plasticity and memory is widely studied and accepted (pubmed: using the keywords (long-term potentiation” and “memory” gives more than 7600 “hits”). The discovery of dendritic spines by Golgi and Cajal was awarded with the Nobel prize in 1906. 
The analysis of dendritic spines is a neuroanatomical readout for  learning and memory processes (pubmed: using the keywords (“dendritic spine” and “learning” gives more than 2500 “hits”). These techniques allow to examine the (sub)cellular mechanisms of learning and memory. Dendritic spines are key structures that facilitate learning and where memory take place and impaired spine dynamics can cause psychiatric and neurodevelopmental disorders (Kasai et al. 2010).
However, the techniques we used here could not be applied in humans. The analysis of dendritic spines might work by using fresh post-mortem brain tissue. However, LTP recording on in-vitro slice preparations could not be performed in humans.

We do not really understand what the reviewer means by
“…., I would appreciate it if you were more explicit about the usefulness of your results and if you did not extrapolate to human. Please remove this information”.
and
“It would be appropriate for them to remove any comments on the applications of their human study and stick to the results.”
As already outlined in our former comments to the reviewers - this is not a study using humans and there are no comments in that manuscript on the applications of our human study, because we have never done a human study in that context. We also have not extrapolate our results to humans.

“I don't know if you have understood the comment I made about statistical analyses. In this sense, although they have used animals, the statistical analyzes are the same. In fact, a very low number of subjects and a high number of comparisons considerably increases the type 1 error. Therefore, it is necessary to apply a correction factor. They must also check if the data conform to normality and, based on this, apply the subsequent statistical analysis”.
We have checked our data for normality using Shapiro-Wilk test and based on that we have now used in several cases the Mann-Whitney test instead of the t-test.
For each experimental setting, we have a certain number of animals (either FTSJ1 knockouts and age-matched littermates). For phenotypic data only single parameters were analyzed in binary comparisons and as such no multiple test correction is applicable. For MS data the reviewer is right that due to the high number of analytes multiple test correction (Bonferroni, Benjamini-Hochberg) would be useful but was not applied because of the rather low number of replicates in this study. In future, material for more replicates has to be stored to allow a more comprehensive record of MS data and in depth statistical analysis.

Reviewer 2 Report

The authors replied to most of the commentary and suggestions of the reviewer. The quality of the manuscript was improved.

Author Response

Thank you very much.